# Atypical E2Fs either Counteract or Cooperate with RB during Tumorigenesis Depending on Tissue Context

**DOI:** 10.3390/cancers13092033

**Published:** 2021-04-23

**Authors:** Eva Moreno, Shusil K. Pandit, Mathilda J. M. Toussaint, Laura Bongiovanni, Liesbeth Harkema, Saskia C. van Essen, Elsbeth A. van Liere, Bart Westendorp, Alain de Bruin

**Affiliations:** 1Department of Biomolecular Health Sciences, Faculty of Veterinary Medicine, Utrecht University, 3584 CT Utrecht, The Netherlands; E.Moreno@uu.nl (E.M.); Shusil.Pandit@med.usc.edu (S.K.P.); touss104@planet.nl (M.J.M.T.); lbongiovanni@unite.it (L.B.); l.harkema@gddiergezondheid.nl (L.H.); s.c.vanessen@uu.nl (S.C.v.E.); e.a.vanliere@uu.nl (E.A.v.L.); B.Westendorp@uu.nl (B.W.); 2Department of Pathology, Keck School of Medicine, University of Southern California, Los Angeles, CA 90033, USA; 3Department of Pediatrics, Division Molecular Genetics, University Medical Center Groningen, University of Groningen, 9217 CP Groningen, The Netherlands

**Keywords:** atypical E2Fs, Rb, interaction, tumorigenesis, transgenic mice

## Abstract

**Simple Summary:**

In virtually all human malignancies, the CDK-RB-E2F pathway is dysregulated resulting in the activation of the E2F transcriptional network. Rb and atypical E2Fs are the most important negative regulators of E2F-dependent transcription during tumorigenesis. However, it is unknown whether they cooporate or act independently in tumor development. Here we show that combined loss of RB and atypical E2Fs in mice enhances tumorigenesis in the liver, while in the pituitary gland, we observe inhibition of tumorigenesis. These findings suggest that the interaction between RB and atypical E2Fs in controlling tumorigenesis occurs in a tissue cell-type specific manner.

**Abstract:**

E2F-transcription factors activate many genes involved in cell cycle progression, DNA repair, and apoptosis. Hence, E2F-dependent transcription must be tightly regulated to prevent tumorigenesis, and therefore metazoan cells possess multiple E2F regulation mechanisms. The best-known is the Retinoblastoma protein (RB), which is mutated in many cancers. Atypical E2Fs (E2F7 and −8) can repress E2F-target gene expression independently of RB and are rarely mutated in cancer. Therefore, they may act as emergency brakes in RB-mutated cells to suppress tumor growth. Currently, it is unknown if and how RB and atypical E2Fs functionally interact *in vivo*. Here, we demonstrate that mice with liver-specific combinatorial deletion of *Rb* and *E2f7/8* have reduced life-spans compared to *E2f7/8* or *Rb* deletion alone. This was associated with increased proliferation and enhanced malignant progression of liver tumors. Hence, atypical repressor E2Fs and RB cooperatively act as tumor suppressors in hepatocytes. In contrast, loss of either *E2f7* or *E2f8* largely prevented the formation of pituitary tumors in *Rb^+/−^* mice. To test whether atypical E2Fs can also function as oncogenes independent of RB loss, we induced long-term overexpression of *E2f7* or *E2f8* in mice. E2F7 and −8 overexpression increased the incidence of tumors in the lungs, but not in other tissues. Collectively, these data show that atypical E2Fs can promote but also inhibit tumorigenesis depending on tissue type and RB status. We propose that the complex interactions between atypical E2Fs and RB on maintenance of genetic stability underlie this context-dependency.

## 1. Introduction

The CDK/RB/E2F axis is the core of G1-S regulation, and thus is essential to ensure proper DNA replication [1]. Upon growth factor signaling, RB gets hyperphosphorylated by Cyclin-CDK complexes and releases the activator E2Fs to induce cell cycle entry. In contrast, the hypophosphorylated form of RB maintains cells in G1 [2]. This notion classifies RB as a cell-proliferation inhibitor. Consequently, mutations affecting RB are frequently encountered in a wide array of human cancers [3,4]. RB binds and inhibits activator E2Fs via a motif known as the pocket domain. Two other proteins, p107 and p130, possess such pocket domains, and like Rb, also appear to regulate cell cycle progression. However, these other two pocket proteins are rarely mutated in tumors [5,6,7,8]. The growth-suppressive properties of RB are thought to be largely dependent on its ability to interact with E2Fs, but also non-canonical functions have been described [9]. RB loss leads to untimely release of activator E2Fs and consequently to upregulation of E2F dependent transcription. This contributes to unscheduled S-phase entry, genomic instability, and eventually cancer progression [10,11].

However, apart from RB (and the related pocket proteins P107 and P130), E2F-dependent transcription is also negatively regulated by repressor E2F family members. In cycling cells, this function is mainly carried out by E2F7 and −8, also known as atypical E2F repressors [12]. Unlike canonical E2Fs, E2F7 and E2F8 lack a pocket protein binding domain and possess an additional DNA binding domain, which suggests that their repressor functions occur independent of RB [13,14,15,16,17]. Moreover, these atypical E2F members redundantly regulate expression of the E2F target genes during late S- and G2 phase, unlike RB/canonical E2Fs which mainly control the G1/S transition [18,19,20]. To date, it is unknown if and how these two seemingly independent E2F-inhibitory mechanisms interact in vivo. In fact, inactivation of RB leads to upregulation of E2F7 and −8 in liver tumorigenesis and cellular senescence [21]. This notion suggests that atypical E2Fs could potentially compensate for the loss of RB. In addition, both RB and atypical E2Fs are thought to play important roles in mediating DNA repair and cell cycle arrest in response to genotoxic reagents [22,23,24,25,26,27,28,29]. Hence, to fully understand how control of E2F-dependent transcription affects tumorigenesis, their genetic interaction must be studied in vivo.

Here, we employed multiple different transgenic and knockout mouse models to study how atypical E2Fs affect tumorigenesis driven by RB loss. We found that loss of atypical E2Fs accelerated the onset of liver tumors but delayed pituitary tumorigenesis in Rb-deficient cells. Moreover, we found enhanced lung tumor formation in transgenic mice with chronic overexpression E2F7 or −8. Thus, atypical E2Fs appear to play a dual role as tumor promoters or protectors and can either compensate or aggravate the tumorigenic effect of RB loss, depending on tissue context.

## 2. Materials and Methods

### 2.1. Animal Experiments

Animal experiments were approved by the Utrecht University Animal Ethics Committee and performed according to institutional and national guidelines. E2f7 and E2f8 knockout mice and doxycycline-inducible transgenic lines were already present in the lab [30,31]. Albumin-Cre mice were derived from Jackson laboratory. Rb conditional knockout mice were provided by Dr. A. Berns (Netherlands Cancer Institute, Amsterdam, The Netherlands). Mice from Figures 1–3 were bred into FVB (Figures 1 and 2: 5th FVB generation, Figure 3: 15th FVB generation) while transgenic mice were maintained on a mixed genetic background 129/Sv (25%) × C57Bl/6 (25%) × FVB (50%) background. Doxycycline (2 g/kg) was administrated ad libitum in pellets to all experimental transgenic mice (Bio Services). BrdU (858811, Sigma-Aldrich) was injected intraperitoneally 2 h prior to euthanasia in doses of 30 µg/g for 4weeks-old mice.

### 2.2. Flow Cytometry

Liver or tumor cell suspensions were prepared from fresh or frozen tissues followed by fixation in 70% ethanol overnight at 4 °C. Cells were washed in PBS and then treated with pepsin (0.5 mg/mL and 0.1N in HCl) to isolate hepatocyte nuclei. Nuclei were stained with anti-BrdU-FITC (Becton Dickinson, 347583, 1:250) and/or propidium iodide (5 µg/mL PI and 250 µg/mL RNAse in PBS) and analyzed with a FACS Calibur flow cytometer (Becton Dickinson) and BD CellQuest Pro software.

### 2.3. Immunohistochemistry and Histological Analysis

Immunohistochemical staining was performed on formalin-fixed and paraffin embedded tissues with a thickness of 4 µm. Endogenous peroxidase activity was blocked with 1% H_2_O_2_. We used 10 mM Citrate buffer (pH 6) for heat-induced antigen retrieval. Sections were stained with standard hematoxylin and eosin (HE). For immunohistochemistry analysis, this involved: anti-Ki67 (Biogenex, MU297-UC; 1:75 in PBS), anti-EGFP (Abcam, Ab6673; 1:800 in PBS), anti ƴH2AX (Cell signaling, S139, 1:500 in PBS), and anti-Caspase 3 (R&D systems, AF835; 1:400 in PBS). Secondary antibodies were biotinylated. Vectastain Elite ABC reagents (Vector Labs) were used according to the manufacturer’s instructions. Slides were counterstained with hematoxylin. Images were taken using a DP25 camera, Labsens soft imaging version 1.1, and Olympus BX45 microscope.

All the HE slides were analyzed by board certified veterinary pathologists (LB, LH and AdB) according to corresponding nomenclature and diagnostic criteria. Quantification of Ki67 in liver slides represents the average of Ki67 positive cells in 5 fields using 40× objective for each condition (non tumor vs. tumor/genotype). Total number of Ki67 positive cells in 5 fields using 40× objective was quantified per lung tumor nodule.

### 2.4. RNA Isolation, cDNA and qPCR

RNA isolation, cDNA synthesis, and quantitative PCR were performed based on manufacturer’s instructions for QIAGEN (RNeasy Kits), Thermo Fisher Scientific (cDNA synthesis Kits), and Bio-Rad (SYBR Green Master Mix), respectively. Reactions were performed in duplicate and a relative amount of cDNA was normalized to house-keeping genes indicated on figure legends using the ΔΔCt method. Sequences of primers used for qPCR are listed in Appendix A.

### 2.5. RNA Sequencing

Total RNA was isolated from livers using the RNeasy Kit (Qiagen, Hilden, Germany). We used 4 livers per condition. To deplete for ribosomal RNA, 5ug of total RNA was used to isolate Poly(A) RNA using poly (A)purist MAG kit (Life Technologies, Carlsbad, CA, USA, AM1922) followed by purification using mRNA only eukaryotic mRNA isolation kit (Illumina, MOE41024). Purified mRNA was used to construct transcriptome libraries using SOLiD total RNA-seq kit (Applied Biosystems Life Technologies, 4445374) using manufacturers’ instructions for low input. Size selected cDNA was amplified, barcoded, and subsequently sequenced using a 5500 W Series Genetic Analyzer (Fisher Scientific, Waltham, MA, USA) to produce 40-bp-long reads. Sequencing reads were mapped against the reference genome (mm9, NCBI37) using BWA (-c –l 25 –k 2 –n 10) software. Gene count tables were then made using FeatureCounts [32]. The R package DESeq2 was used to call differentially expressed genes between controls and knock-out tumor samples with a Benjamini–Hochberg-adjusted *p* value of less than 0.05 [33]. Heatmaps were generated using the R package pheatmap. GEO accession number: GSE172508. 

### 2.6. Statistics

The number of mice and the specific statistical tests for each figure are indicated in the legends. All statistical analyses were performed using SPSS statistical package or SigmaPlot 13.0 software. Asterisks indicate where significant differences were seen. Where relevant for understanding the figure and individual comparisons, we indicate with “n.s.” that significance was not reached.

## 3. Results

### 3.1. RB and Atypical E2F Cooperate to Prevent Liver Cancer

Previous studies showed that loss of RB enhanced tumor formation in livers exposed to the hepatocarcinogen diethylnitrosamine (DEN) [34]. Those tumors expressed high levels of RB/E2F target genes, indicating that tumorigenesis in RB deficient livers may result from the deregulation of E2F dependent transcription. Atypical E2Fs are downstream targets of RB and their expression is frequently increased in tumors. Paradoxically, atypical E2Fs act as tumor suppressors in the liver [10,31]. Therefore, we asked whether atypical E2Fs could at least in part compensate for the loss of RB, via repression of E2F-responsive genes (Figure 1A). Thus, deletion of E2f7 and −8 would exacerbate the formation of Rb-deficient liver tumors. To test this hypothesis, we conditionally deleted E2f7, E2f8, and Rb (*Alb-Rb^−/−^7^−/−^8^−/−^*) in murine livers using the Cre/LoxP transgenic approach under the control of the hepatocyte-specific albumin promoter (Alb-Cre). We then compared lifespans and spontaneous liver tumor incidence of these triple knockout mice with E2f7 and E2f8 double knockout (*Alb-7^−/−^8^−/−^*), Rb single knockout (*Alb-Rb^−/−^*), and control littermates (Appendix A). Mice were either found dead, or were euthanized when they were moribund. Deletion of atypical E2fs or Rb alone did not significantly alter the lifespan, but *Alb-Rb^−/−^7^−/−^8^−/−^* mice, namely males, lived significantly shorter lives than control mice, *Alb-Rb^−/−^* mice, and *Alb-7^−/−^8^−/−^* mice (Figure 1B and Appendix A). Post-mortem analysis of the livers showed that 65% of *Alb-Rb^−/−^7^−/−^8^−/−^* mice had liver tumors, whereas only one Rb^−/−^ single knockout mouse carried a liver tumor (Figure 1C). There was a strong gender bias in liver tumor incidence, because nearly all tumors were seen in males (Appendix A). This suggests that male *Alb-Rb^−/−^7^−/−^8^−/−^* mice develop tumors earlier than *Alb-7^−/−^8^−/−^* mice. Indeed, *Alb-7^−/−^8^−/−^* mice also developed liver tumors at the same incidence rate, however the tumor latency was longer compared to *Alb-Rb^−/−^7^−/−^8^−/−^* (Figure 1C and Appendix A). These findings indicate that additional deletion of RB in E2F7/8 deficient liver has no major impact on tumor initiation, but results in enhanced tumor progression. Histopathology analysis revealed that *Alb-Rb^−/−^7^−/−^8^−/−^* mice had an increased incidence of malignant liver tumors (hepatocellular carcinomas) compared to *Alb-7^−/−^8^−/−^* mice, although this difference did not reach statistical significance (Figure 1D). Most HCCs were well differentiated, but 10% of those HCCs in *Alb-Rb^−/−^7^−/−^8^−/−^* mice were poorly-differentiated, suggesting a faster progression. Body and liver weights did not significantly differ between *Alb-7^−/−^8^−/−^* and *Alb-Rb^−/−^7^−/−^8^−/−^* mice (Appendix A). No other pathologies were observed in the livers of the male *Alb-Rb^−/−^7^−/−^8^−/−^* mice, suggesting that their decreased lifespan was directly related to the liver tumors seen in these mice. Together, these data show that atypical E2Fs cooperate with RB to suppress liver cancer and that their combined loss causes liver cancer-associated mortality.

### 3.2. Loss of RB Results in Enhanced Proliferation and Deregulation of Cell Cycle Control in Atypical E2F-Deficient Liver Tumors

We performed immunohistochemical analysis of Ki67 expression in the tumors and the adjacent normal liver tissues. This analysis revealed a significant increase in percentage of Ki67 positive hepatocytes in *Alb-Rb^−/−^7^−/−^8^−/−^* and *Alb-7^−/−^8^−/−^* tumors compared to adjacent non-tumor tissue (Figure 2A,B). Furthermore, *Alb-Rb^−/−^7^−/−^8^−/−^* liver tumors showed significantly higher levels of Ki67-positive cells than those from *Alb-7^−/−^8^−/−^* mice, indicating that additional deletion of Rb enhanced proliferation of tumor cells.

The growth-suppressive properties of RB are thought to be largely dependent on its ability to interact with E2F1/2/3 to regulate E2F-dependent transcription, although non-canonical functions have been described [9]. Importantly, we showed previously that the main tumor suppressive mechanism of atypical E2Fs is their repressive function on E2F-dependent transcription [10,31]. We therefore investigated whether the enhanced proliferation and tumorigenesis in the triple-knockout mice could be explained by further deregulation of E2F dependent transcription in those livers. Firstly, we analyzed proliferation and E2F target gene expression in juvenile livers, when tumors are not yet present and the proliferation rate of hepatocytes is high. The DNA replication rates, shown by BrdU incorporation, were increased in *Alb-Rb^−/−^7^−/−^8^−/−^* compared to *Alb-7^−/−^8^−/−^* livers (Appendix A). However, a similar increase was seen when comparing the Cre-negative littermates of *Alb-Rb^−/−^7^−/−^8^−/−^* livers to *Alb-7^−/−^8^−/−^* livers, indicating that additional deletion of Rb was not responsible for an increase in DNA replication at 4 weeks of age (Appendix A). Consistently, E2F target gene expression was not increased in juvenile *Alb-Rb^−/−^7^−/−^8^−/−^* livers compared to *Alb-7^−/−^8^−/−^* livers (Appendix A). We then analyzed the expression of multiple E2F-dependent genes in tumor and adjacent tumor areas. Consistent with our observations in the juvenile mice, there was not a significant up-regulation in *Alb-Rb^−/−^7^−/−^8^−/−^* tumors compared to *Alb-7^−/−^8^−/−^* tumors, with the exception of the licensing factor Cdt1 (Figure 2C). To explore gene expression changes in an unbiased manner, we performed RNA sequencing analysis on non-tumor tissue from controls and *Alb-7^−/−^8^−/−^* and *Alb-Rb^−/−^7^−/−^8^−/−^* tumors. Pathway analysis of differentially expressed genes in the tumor samples revealed up-regulation of genes associated with cell cycle and cancer-related signaling pathways in tumors of both genotypes compared to wild-type control livers(Appendix A). Downregulated pathways were mostly involved in energy metabolism (Appendix A). Strikingly, we only identified 44 genes which were significantly differentially expressed between *Alb-Rb^−/−^7^−/−^8^−/−^* and *Alb-7^−/−^8^−/−^* tumors compared to controls, indicating that these tumors are remarkably similar (Appendix A). We then plotted expression of a panel of well-described E2F target genes. This analysis did not reveal differences between the tumor samples from *Alb-Rb^−/−^7^−/−^8^−/−^* and *Alb-7^−/−^8^−/−^* livers, consistent with the qPCR data shown in (Appendix A). Together, these data indicate that the cooperation between RB and atypical E2Fs in tumor suppression extends beyond compensatory E2F target gene repression.

Next, we determined whether the combined deletion of Rb/E2f7/8 changed the polyploidization status and cell cycle profile of hepatocytes, since these cell cycle genes have been shown to control liver cell polyploidization and cell cycle progression. Previous studies demonstrated that loss of RB enhances liver cell polyploidization, whereas ablation of E2F7/8 prevents polyploidization [34,35]. Consistent with previous studies, our *Alb-7^−/−^8^−/−^* juvenile livers showed a marked reduction of 4C and 8C nuclei compared to control littermates (Appendix A). Remarkably, we found a similar ploidy reduction in juvenile *Alb-Rb^−/−^7^−/−^8^−/−^* triple knockout hepatocyte nuclei (Appendix A), indicating that atypical E2Fs counteract RB in regulating liver cell polyploidization. We then analyzed the ploidy status in liver tumors and their adjacent normal tissue. In line with the observations in the juvenile mice, *Alb-7^−/−^8^−/−^* and *Alb-Rb^−/−^7^−/−^8^−/−^* hepatocytes remained mostly diploid in non-tumor areas. Interestingly, in liver tumors, we observed an altered cell cycle profile marked by an increase of 4C cell populations in *Alb-Rb^−/−^7^−/−^8^−/−^* tumors compared to *Alb-7^−/−^8^−/−^* tumors (Figure 2D,E). Tetraploidization or presence of G2-like cells could represent an early event in the tumorigenesis of *Alb-Rb^−/−^7^−/−^8^−/−^* livers, indicating enhanced genomic instability. To explore whether genomic instability was enhanced in *Alb-Rb^−/−^7^−/−^8^−/−^* tumors, we plotted a gene signature related to chromosomal instability [36]. This signature contains many genes involved in G2/M progression. Although expression of this signature was increased in most tumors compared to wild type livers, we did not observe a difference between *Alb-Rb^−/−^7^−/−^8^−/−^* and *Alb-7^−/−^8^−/−^* tumors (Appendix A), consistent with our notion that the emergence of 4C cells had been an early event. In line with this, DNA damage, measured by quantification of ƴH2AX-postive hepatocytes, was not different between *Alb-Rb^−/−^7^−/−^8^−/−^* and *Alb- 7^−/−^8^−/−^* tumors (Appendix A).

Previous studies have shown that deletion of Rb in livers can cause defects in mitotic entry and activation of the DNA damage checkpoint [37]. Furthermore, the E2F7/8 loss can cause defects in the DNA damage checkpoint [38]. Thus, combined loss of RB and atypical E2Fs could cause altered cell cycle fates of damaged cells to eventually promote tumorigenesis. 

Together, our data demonstrate that atypical E2Fs act as tumor suppressors and negative regulators of proliferation in RB-null livers. Although they are active during different phases of the cell cycle, Rb during G1/S transition and atypical E2Fs during S/G2 (Figure 1A), their combined action could be required to maintain genomic integrity and prevent oncogenesis in the liver.

### 3.3. E2F7 and −8 Promote Tumorigenesis in Pituitary Gland of Rb^+/−^ Mice

We next sought to investigate if the genetic interaction between RB and atypical E2Fs would be different in other tissues. To this end, we investigated the consequences of E2F7 and E2F8 loss in Rb^+/−^ mice, a well-established loss-of-heterozygosity model characterized by the emergence of tumors in the pituitary and thyroid glands. Because global combined deletion of E2f7 and E2f8 results in embryonic lethality due to a placental phenotype [30,39], we crossed conventional knock-out mice with single homozygous deletion of E2f7 or E2f8 with Rb^+/−^ mice to generate the following experimental cohorts: *Rb^+/−^*; *Rb^+/−^7 ^−/−^*; *Rb^+/−^8 ^−/−^*; *7^−/−^* and *8^−/−^* mice. We performed a cross-sectional pathology study at the age of 14 months. Consistent with previous reports, a high percentage of adult Rb^+/−^ mice developed spontaneous pituitary tumors. Surprisingly, additional deletion of E2f7 or E2f8 resulted in significantly lower pituitary tumor incidence compared to Rb^+/−^ mice (Figure 3A). Mice with single deletion of E2f7 or −8 did not develop any spontaneous pituitary tumors.

We further examined those pituitary tumors by performing pathological analysis. Histology showed that the pituitary glands of single E2f7 (*7^−/−^*) or E2f8 (*8^−/−^*) mutant littermates appeared completely normal (Appendix A). The histological features of *Rb^+/−^* pituitary tumors were entirely consistent with previous studies where carcinomas originated from the intermediate lobe were highly prevalent [40] (Appendix A). The percentage of malignant tumors was reduced in *Rb^+/−^ 7^−/−^* and *Rb^+/−^ 8^−/−^* compared to *Rb^+/−^* mice (Figure 3B,C). These results strongly suggest that E2F7 and −8 might act as oncogenes in Rb-deficient pituitaries.

Beside pituitary tumors, *Rb^+/−^* mice are also predisposed to develop c-cell hyperplasia, which subsequently can progress into medullary thyroid carcinomas [41,42]. We therefore also histologically examined the incidence of hyperplasia and tumorigenesis in the thyroid gland and other glands, namely adrenal and mammary glands. We observed a slight trend, although not a significant one, of increased thyroid tumor incidence in *Rb^+/−^7^−/−^* compared to *Rb^+/−^* mice (Figure 3D). The opposite trend occurred in *Rb^+/−^8^−/−^* mice, where loss of E2F8 reduced the incidence of Rb-deficient thyroid tumors. Pathological analysis revealed that four out of the five *Rb^+/−^7^−/−^* tumor lesions were medullary thyroid carcinomas, suggesting that loss of E2F7 aggravates the malignant progression of Rb-deficient thyroid tumors (Figure 3E,F). We also analyzed the adrenals and mammary glands of these mice, but we only incidentally observed tumors, with no statistically increased incidence in any of the analyzed genotypes (Figure 3D). Early lesions (hyperplasia) were frequently encountered but there were no significant differences between genotypes (Appendix A). Overall, these data demonstrate that E2F7 and E2F8 clearly contribute to the tumor formation in the pituitary glands of *Rb^+/−^* mice, while in the thyroid glands, loss of atypical E2Fs does not have a significant impact on RB loss dependent tumorigenesis. This demonstrates that the interaction between atypical E2Fs and RB is highly tissue-specific.

### 3.4. E2F7 and −8 Overexpression Promotes Spontaneous Lung Tumorigenesis

Having discovered that atypical E2Fs can have oncogenic functions in a Rb-mutant background, we asked if overexpression of E2F7 or −8 by itself could be sufficient to drive tumorigenesis. Previously we developed E2f7 and E2f8 transgenic (Tg) mice capable of blocking the proliferation of chemically-induced hepatocellular carcinomas, consistent with their role as tumor suppressors in the liver [31]. These mice carry ubiquitously expressed doxycycline-inducible E2f7/8 alleles. We now induced long-term overexpression of atypical E2Fs and analyzed tumor incidence in aged mice. We maintained a cohort of male and female *control, E2f7, E2f8*, and *E2f8^DBDmut^ Tg* mice on lifelong doxycycline chow, starting from young adulthood- 3-4 months old (Figure 4A). *E2f8^DBDmut^* mice carrying mutations in both DNA-binding domains were included as an additional control group because they are transcriptionally inactive [31]. Chronic E2F7 or E2F8 overexpression did not significantly affect lifespan, although we noticed that a subset of male and female *E2f7* Tg mice died within the first weeks of doxycycline induction (Figure 4B and Appendix A).

Immunohistochemistry staining showed that transgenic E2F7-EGFP and E2F8-EGFP were mainly found in proliferative adult tissues (Appendix A). This is consistent with previous findings describing that E2F7/8 proteins are efficiently degraded in non-cycling cells [10,20]. We also observed a decline in EGFP expression over time (Appendix A). This decrease in transgene expression can be explained by a decrease in overall proliferation rates during aging, but also by the negative selection pressure on transgenic cells caused by the antiproliferative effect of atypical E2Fs. Furthermore, we found increased apoptosis in small intestinal crypts of *E2f7* Tg mice treated for 3 months with doxycycline (Appendix A), which could also contribute to the lower number EGFP positive cells observed in aged Tg mice. Thus, inhibition of cell proliferation in intestinal epithelium and other rapidly dividing tissues, could at least partially explain why some *E2f7* Tg mice presented early lethality (Figure 4B and Appendix A).

To initially screen for tumors, we performed necropsy on *control*, *E2f7*, *E2f8*, and *E2f8^DBDmut^* Tg mice. Interestingly, we observed an increased presence of tumors in the lungs of transgenic mice, but not in any other organ (Appendix A). To confirm and quantify this observation, we performed microscopic analysis of at least 5 different lung sections per mouse. Both *E2f7* and *E2f8* Tg mice indeed presented a higher microscopic lung tumor incidence than controls and *E2f8^DBDmut^* Tg (Figure 4C and Appendix A). Lung tumors are often observed in ageing mice, especially in the FVB strain [43], and these findings suggest that overexpression of atypical E2Fs may drive oncogenesis in some tumor-prone tissue contexts. Histopathology revealed that the vast majority of the tumors in all genotypes were focal adenomas with rather low proliferation rates (Appendix A). Nonetheless, we found a few cases of more aggressive multifocal adenocarcinomas in *E2f7* and *E2f8* Tg mice, and not in the other genotypes shown in (Figure 4E,F). Importantly, *E2f8^DBDmut^* Tg mice, which carry mutations in the DNA binding domain of E2f8, did not show increased lung tumor incidence, which strongly suggests that the oncogenic action of atypical E2Fs in the lungs is due to mis-regulation of target gene transcription rather than due to another mechanism.

This raises the question of whether the E2F target gene expression was repressed or activated in the lung epithelium. To answer this, we analyzed target gene expression after 3 days of E2F7/8 induction in 21 year-old juvenile Tg mice for 3 days as previously described [31]. Doxycycline caused an induction of *E2f7* Tg and *E2f8* Tg proteins, which was analyzed by EGFP-IHC staining, and mild repression of E2F target transcripts (Appendix A). These results show that long term in vivo overexpression of atypical E2Fs can promote tumor growth in the lungs, most likely via a mechanism involving transcriptional repression.

## 4. Discussion

The results presented here highlight the complexity of the E2F/RB pathway in tumorigenesis. Unrestrained E2F-dependent transcription is linked to tumor progression and poor prognosis in several cancer types. However, the consequences of deleting RB and atypical E2Fs—two independent mechanisms that both repress E2F-dependent transcription—were strikingly context and tissue-specific. While atypical E2Fs and RB cooperate to prevent liver tumorigenesis, they counteract each other in pituitary glands. In line with this, we revealed that long-term expression of atypical E2F enhances spontaneous lung tumorigenesis. This suggests that E2F7/8 can act as oncogenes besides their tumor suppressor functions.

Although surprising, our findings are consistent with previous studies that also revealed dual roles for other E2F family members. For example, various mouse models unequivocally showed that E2f1 and E2f3 can both act as tumor suppressors or oncogenes, depending on tissue context [10,41]. But what determines the tissue specific functions of atypical E2Fs? In chemical-induced skin carcinogenesis, we previously showed that atypical E2Fs block tumorigenesis by mediating a cell cycle arrest in cells with unresolved DNA damage [11]. In the genetic models presented here, levels of genotoxic stress are much lower and possibly other downstream effects of atypical E2Fs come into play. In the liver, the maintenance of cell cycle checkpoints appears to be critical herein. We demonstrated previously that E2F7 and −8 suppress tumor growth in the liver via transcriptional repression of E2F targets and possibly through promoting polyploidization [10,31]. Although RB loss in the liver results in elevated expression of the same target genes [22], the timing during the cell cycle is completely different [1,19,44]. RB prevents cells from making the G1-S transition, whereas atypical E2Fs repress target genes during late S and G2. Hence atypical E2Fs are involved in the DNA damage checkpoint [18,45], whereas RB activates the G1/S checkpoint [1,46]. Manipulating both checkpoints by combined deletion of E2F7/8 and RB would thus further enhance the risk of genomic instability and cancer than either one of these interventions alone.

This checkpoint-based hypothesis does not explain our observations that E2f7 or E2f8 deletion prevented pituitary tumor formation in the Rb^+/−^ model. It should be kept in mind this is a different tumor model, in which loss of heterozygosity (LOH) of the Rb gene is a key early event in the formation of pituitary tumors, whereas Rb is homozygously deleted in our Alb-Cre model. Therefore, it is possible that atypical E2Fs promote LOH in the pituitary tissue. Although it is difficult to experimentally address this possibility, previous studies showed that inhibition of DNA repair genes could cause LOH [47]. This fits with our previous observations that E2F7 and −8 repress multiple DNA repair factors during G2 [18]. However, it is not clear why the pituitary tissue would be so exquisitely sensitive to E2F7/8-induced LOH of the Rb locus. No other tissues in the Rb^+/−^ mice showed evidence that E2F7/8 loss promoted LOH.

Our long-term E2F7/8 overexpression studies suggest that lung is another tissue in which atypical E2Fs can act as oncogenes in vivo. This is consistent with previous in vitro work claiming an oncogenic function of E2F8 in lung cancer cell lines [48]. The question then arises if LOH plays a role in the emergence of the lung adenomas and adenocarcinomas in the *E2f7/8* Tg mice. Our Tg mice were bred one generation into the FVB background. FVB mice are more prone to develop these spontaneous lung tumors than other strains, suggesting a genetic predisposition [43]. Possibly, FVB mice first acquire a heterozygous mutation in a key lung tumor suppressor gene, and LOH is an important second hit event for these tumors to appear. Therefore, LOH in this model is difficult to prove but again conceivable. Nevertheless, cell non-autonomous effects could also play a role in lung tumorigenesis. We reported before that *E2f7/8* Tg mice inhibits proliferation of rapidly dividing tissues. This could include white blood cells, and immunosurveillance of (pre-)malignant cells in the lungs could thus be impaired in *E2f7/8* Tg mice.

## 5. Conclusions

Taken together, the in vivo studies presented here indicate that atypical E2Fs and RB both control multiple mechanisms that are required to maintain genomic integrity. The importance of balancing E2F-dependent transcription extends far beyond controlling the G1/S transition and apoptosis. Its role in maintenance of genomic stability is highly complex and tissue-dependent. Moreover, we showed previously evidence that E2F-dependent transcription can be highly heterogeneous between single cancer cells [38]. Therefore, it will be important to study how variation in E2F-dependent transcription in both RB-mutant and RB-intact tumors will affect for example prognosis and anti-cancer drug responses.

## Figures and Tables

**Figure 1 cancers-13-02033-f001:**
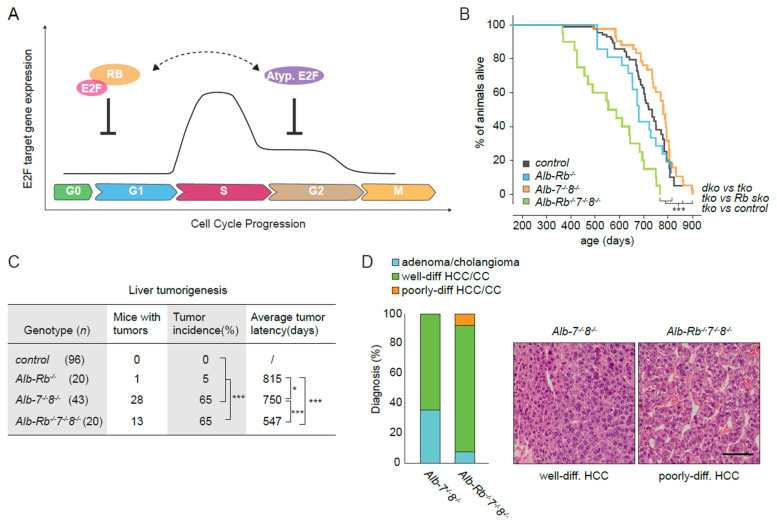
Atypical E2Fs and RB cooperate to prevent liver cancer. (**A**) Schematic representation of the regulation of E2F dependent transcription by RB and atypical E2Fs in the cell cycle. (**B**) Kaplan–Meier overall survival curves of control (*n* = 96), *Alb-Rb^−/−^* (Rb sko; *n* = 20), *Alb-7^−/−^ 8^− /−^* (dko; *n* = 43) and *Alb-Rb^−/−^ 7^−/−^ 8^−/−^* (tko; *n* = 20) mice. (**C**) Table indicating tumor incidence (%) and tumor latency (days) of the indicated genotypes at the end of life-span. (**D**) Histological classification of tumors from *Alb-7^−/−^ 8^− /−^* (*n* = 28) and *Alb-Rb^−/−^ 7^−/−^ 8^−/−^* (*n* = 13) mice. In the right representative pictures of well and poorly differentiated HCC tumors. Scale bar: 50 µm. Data information: In (**B**,**C**); * *p* < 0.05, *** *p* < 0.001 (Log Rank Mantel-Cox test and Chi-square (tumor incidence)).

**Figure 2 cancers-13-02033-f002:**
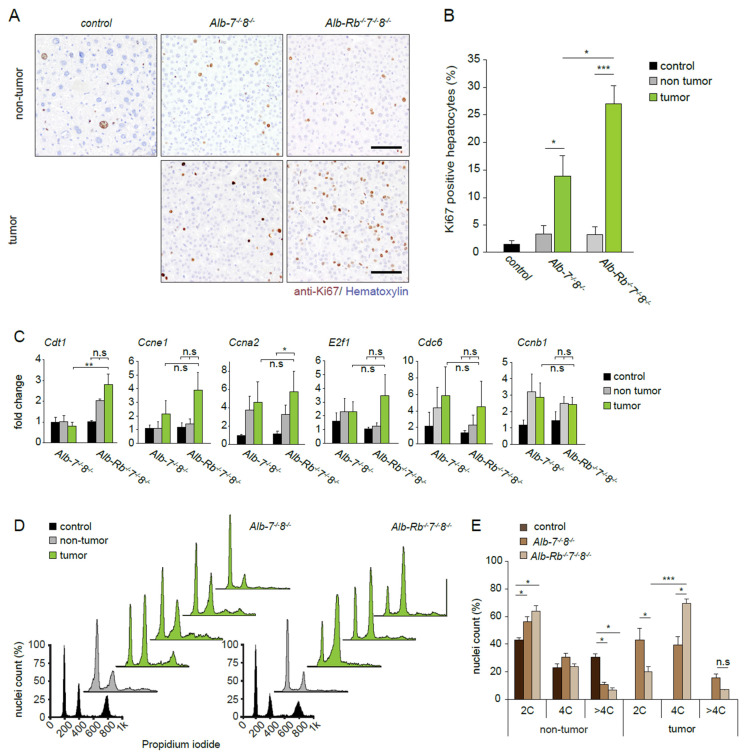
Combined loss of E2F7/8 and RB results in rapidly proliferating liver tumors. (**A**) Representative pictures of Ki67 showing proliferating cells in non-tumor and tumor areas of livers from the indicated genotypes. Tumors were collected at the end of life-span. Scale bars: 50 µm. (**B**) Quantification of Ki67 immunohistochemistry in 5 fields (40× objective) in tumor and adjacent non tumor areas of the livers from the indicated genotypes. Controls represent Alb-Cre negative littermates analyzed at the same time point as Alb-Cre positive ones. The data are presented as average ±SEM (1 tumor per mice was analyzed/mouse; *n* = 6–7 mice). (**C**) Transcript levels of E2F target genes in the indicated areas from *Alb−7^−^/^−^8^−^/^−^* and *Alb-Rb^−^/^−^7^−^/^−^8^−^/^−^* mice. Fold changes were adjusted to average of controls and *Actb* and *Gapdh* were used to normalize the expression. Data represent average ± SEM (*n* = 5–6 mice). (**D**) Representative cell cycle profiles analyzed by flow cytometry of the indicated areas and genotypes. E. Quantification of nuclei from the cell cycle profiles in (**D**) Data represent average ± SEM (*n* = 3–5). Data information: In (**B**) * *p* < 0.05, (Mann–Whitney Rank Sum test); (**C**,**E**) * *p* < 0.05, ** *p* < 0.01, *** *p* < 0.001 (One Way ANOVA).

**Figure 3 cancers-13-02033-f003:**
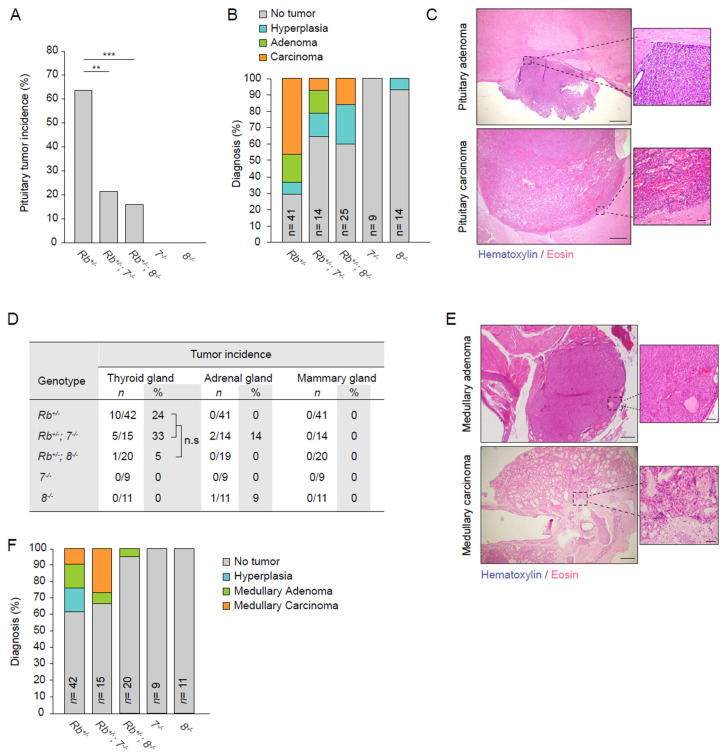
E2F7 and −8 promote tumorigenesis in pituitary glands with RB loss (**A**) Pituitary tumor incidence (%) in 14 months old mice from the indicated genotypes. Data collected from *Rb^+^/^−^ n* = 41; *Rb^+^/^−^, 7^−^/^−^ n* = 14; *Rb^+^/^−^, 8^−^/^−^ n* = 25; *7^−^/^−^ n* = 9; *8^−^/^−^ n* = 14. (**B**) Histological diagnoses of pituitary glands from (**A**,**C**) Representative images of histological features of pituitary tumors. Scale bars: left picture 500 µm; right 50 µm. (**D**) Microscopic tumor incidence of additional neuroendocrine glands from mice in (**A**,**E**) Representative images of histological features of thyroid tumors. Scale bars: left picture 500 µm; right 50 µm. (**F**). Histological diagnoses of thyroid tumors from (**E**) Data information: In (**A**,**D**); n.s not significant, ** *p* < 0.01, *** *p* < 0.001 (chi-square).

**Figure 4 cancers-13-02033-f004:**
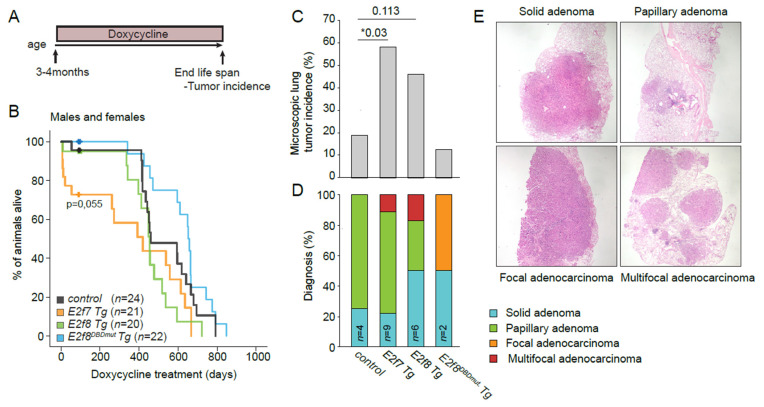
E2F7 or −8 overexpression promotes spontaneous lung tumorigenesis. (**A**) Experimental scheme of the long-term induction of E2F7 and −8 in the transgenic mice. (**B**) Kaplan–Meier overall survival of males and females control (*n* = 24); *E2f7* Tg (*n* = 21); *E2f8* Tg (*n* = 20); *E2f8DBD^mut^* Tg (*n* = 22) mice. (**C**) Microscopic lung tumor incidence in the indicated genotypes of mice euthanized at the end of life-span. *Control* (*n* = 16); *E2f7* Tg (*n* = 12); *E2f8* Tg (*n* = 13); *E2f8DBD^mut^* Tg (*n* = 16) mice. (**D**) Pathological analysis of the lung tumors. (**E**) Representative images of the different pathological diagnoses in (**E**) Data information: In (**B**) Log-rank Mantel–Cox test. In (**C**,**D**) n.s not significant, * *p* < 0.05 (chi-square).

## Data Availability

All RNA-sequencing data were deposited in the Gene Expression Omnibus with the following accession number GSE172508.

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
