# Peer review of "Atypical E2Fs either Counteract or Cooperate with RB during Tumorigenesis Depending on Tissue Context"

_cancers, 2021, doi:10.3390/cancers13092033_

Round 1

Reviewer 1 Report

This study investigated the genetic interaction of E2f7 and E2f8 with Rb by using transgenic mice. By studying various tissues, roles of E2f7 and E2f8 were found to be counteracting or cooperating with Rb. Findings are solid although it is currently known about why there are opposing roles in E2f7 and E2f8, which make this current study be a valuable foundation for future in depth research. Results are presented clearly with appropriate statistics. I have some comments which may help further improve the manuscript:

  1. On line 48, please explain more about the pocket domain, which would help readers to have better understanding of the background.
  2. In Figure 1B, please define the meanings of dko, tko and Rb sko.
  3. In Figure 1C, the tumour incidence rate is similar between Alb-7-/-8-/- mice and Alb-Rb-/-7-/-8-/- mice. It would be interesting to compare the tumour size in mice of the same age. Also it would be informative to see if there is any association between Ki67 positive cells and the tumour size.
  4. On line 199, the results only show that ploidy reduction were observed in both Alb-7-/-8-/- and Alb-Rb-/-7-/-8-/- livers, which cannot justify the statement of “that RB might be dependent on atypical E2Fs in regulating liver cell polyploidization”
  5. For comparison between Alb-7-/-8-/- mice and Alb-Rb-/-7-/-8-/- mice, it would be informative to discuss why the enhanced proliferation and cell cycle control deregulation in Alb-Rb-/-7-/-8-/- mice did not worsen the tumour incidence.
  6. On line 297, please explain the meaning of E2f8 DBDMut at its first appearance.
  7. Seems the conclusion is missing on line 404.

Reviewer 2 Report

In this manuscript, Moreno et al. address how Retinoblastoma protein (RB) and atypical E2F members E2F7 and E2F8 functionally interact in vivo during tumorigenesis. The topic is interesting and the in vivo experiments making use of different transgenic and knockout models are overall well performed. However, the results are mostly descriptive, and little mechanistic insight is provided to support their conclusions. The manuscript in its present form lacks a clear focus and is too preliminary. Therefore, before publication can be recommended, several aspects should be developed in more detail and a number of points should be clarified or corrected. Specific points that deserve special attention are outlined below:

-        Given the well-established role of either RB or atypical E2Fs as negative regulators of E2F target gene expression it is surprising that the concomitant loss of all three genes had such a small impact in gene expression in liver tumors (only Cdt1 expression was significantly elevated on TKO mice). This seems to be partly due to a substantial sample variability, as shown in figure 2C. The number of analyzed mice should be increased to reduce this variability and help identify specific genes that are behind the phenotypes described by authors.

-        Remarkably, despite having examined only 5 genes, the authors conclude that “cooperation between RB and atypical E2Fs in tumor suppression extends beyond compensatory E2F target gene repression”. However, the number of genes that were analyzed was too short to make such a claim. The authors need to broaden the range of analyzed E2F/RB target genes to include a more complete panel with those involved in various cell cycle phases, DNA replication and repair, cytokinesis etc. to begin to define the mechanism underlying liver tumorigenesis in E2F7/E2F8 knockout mice and the impact of RB loss in this context. In this regard, a transcriptomic analysis would be highly informative.

-        The expression of E2F1 is increased in juvenile Alb-Rb-/-7-/-8-/- and Alb-7-/-8-/- livers, but not in adult non-tumor samples, nor in DKO tumor samples, pointing to complex regulatory effects. Based on this data, a more extensive gene expression analysis in juvenile livers, before tumor onset, would help understand the mechanism of tumor development in this model.

-        Authors claim that in juvenile mice DNA replication rates are increased in Alb-Rb-/-7-/-8-/- TKO compared to Alb-7-/-8-/- DKO livers. (Figure S2A). However, in control (Cre-negative) Alb-Rb-/-7-/-8-/- mice BrdU incorporation is also significantly higher compared to Alb-7-/-8-/- control (Cre-negative) juvenile livers. How do authors explain this result? More importantly no differences can be observed in DNA replication rates between Cre-negative and Cre-positive Alb-Rb-/-7-/-8-/- mice. Based on these data, it is incorrect to conclude that DNA replication rates are increased in TKO juvenile livers. Authors should clarify this point.

-        In liver tumors authors observed an increase of a 4C cell population in Alb-Rb-/-7-/-8-/- tumors compared to Alb-7-/-8-/- tumors. This is not surprising, since it has already been shown that RB loss leads to liver tumors harboring tetraploid genomes associated with a chromosomal instability gene expression signature (ref 26). However, the claim that tetraploidization or presence of G2-like cells could represent an early event in the tumorigenesis of TKO livers is not substantiated by any data. Authors need to characterize further this 4C population in DKO and TKO liver tumors. Is the expression of cytokinetic or checkpoint genes deregulated in this context? Do these cells accumulate DNA damage and signs of chromosome instability? Are there signs of replication stress? Are mitotic markers affected? These experiments will help elucidate the functional cooperation of RB and E2F7/8 in liver tumorigenesis at the molecular level.

-        Crossing Rb+/- mice, a well-established loss-of-heterozygosity model characterized by emergence of tumors in the pituitary and thyroid glands, with E2F7-/- or E2F8-/- mice, authors claim that E2F7 and E2F8 contribute to the tumor formation in the pituitary glands of Rb+/- mice, but not in the thyroid glands. Results in this section are too preliminary. Authors do not show any data to address why this oncogenic role of E2F7 and E2F8 takes place in the pituitary gland and not in the thyroid gland. Moreover, a comparison of this model with the Alb-Rb-/-7-/-8-/- TKO is not pertinent in the absence of more data. In the Rb+/- E2F7-/- or Rb+/- E2F8-/- context, complex compensatory mechanisms might be taking place by the presence of the other atypical E2F. Again, an extensive gene expression analysis is needed to explore the different behavior of atypical E2Fs in different contexts. In this line, authors could consider analyzing the tumorigenic potential in liver of Alb-Rb-/-7-/- and Alb-Rb-/-8-/- DKO mice.

-        Similarly, the section on E2f7 and E2f8 transgenic (Tg) mice is too preliminary. Of note, tumor incidence upon E2F8 overexpression is not significant, thus, making this result quite weak. Moreover, authors state that the oncogenic role of E2F7 and E2F8 in the lungs most likely involves transcriptional repression of E2F target genes. Authors reach this conclusion by analyzing only the expression of only 2 target genes: E2F1 and CDC6. Strikingly, while E2F1 expression is repressed by E2F7 but not E2F8, CDC6 expression is repressed to the same extent by E2F7, E2F8 and the mutant E2F8 lacking the DNA binding domain. These data are quite unsatisfactory, and authors need to extend their analyses to strengthen their conclusions, otherwise the data obtained with the transgenic mice should not be included in the manuscript.

-        In the Materials and Methods section the sequence of the primers used for qPCR should be provided.

-        In Figure S2C, what samples are being compared for the statistical significance test?

-        Some references need to be corrected.

Reviewer 3 Report

This manuscript examines the impact of the atypical E2F family members E2F7 and E2F8 on cancer development in both loss and gain of function mouse models. Interestingly, E2F7 and 8, like some other E2F family members, are shown to have either tumor suppressive or oncogenic properties depending on the tissue and experimental condition. Specifically, loss of E2F7/8 cooperates with the loss of RB to promote DEN-induced liver carcinogenesis while E2F7/8 loss protects RB +/- mice from developing spontaneous pituitary tumors. Whole body over expression of either E2F7 or 8 is also shown to specifically predispose RB+/+ mice to lung tumorigenesis. A strength of this manuscript are the use of genetically engineered mouse models and the different experimental systems used to study the cancer-modulatory properties of E2F7 and 8 and their interactions with RB.  The major weakness is the lack of mechanistic data to explain how E2F7 and 8 can either suppress or promote tumor development dependent on the tissue and experimental conditions. Nonetheless, this study adds to our understanding of the complex roles that the E2F family of transcription factors plays in cancer. Minor points:

1) line 54: the authors state that "atypical E2F members mediate expression of E2F target genes during late S- and G2 phase". Shouldn't it be that atypical E2Fs "repress" target genes or  "regulate" target genes in late S and G2?

2) line 60: the authors state that both RB and atypical E2Fs are thought to play important roles in mediating DNA repair and cell cycle arrest. This is true but the authors only reference papers where RB and E2F7/8 mediate genome stability and cell cycle arrest. 

3) line 297: The E2F8 DBD mutant is a nice control but it is not explained that it is a DNA binding mutant until later on line 326. This should be explained the first time it is introduced.

4) line 388: The authors suggest that FVB mice carry a heterozygous mutation in a key lung tumor suppressor gene. I may be missing something but by definition, but mouse strains like FVB should be homozygous at all loci. 

5) The authors mention their previous study using E2F7 and E2F8 null mice in a skin carcinogenesis model. How do those previous findings in the skin compare with the new studies presented here in the liver, pituitary and lung? Can the authors provide a more comprehensive understanding of the functions of E2F7 and 8 by including this and perhaps other studies in other tissues and under different conditions? 

Round 2

Reviewer 2 Report

Overall, I believe the manuscript under consideration can be accepted for publication in the Cancers journal after the authors properly addressed my points. The new data are presented and discussed in the text in a convincing manner.